# The Nurses' Innovative Behavior Inventory (NIBI): A development and validation study

Elham Shahidi Delshad[1], Mohsen Soleimani[2,3], Armin Zareiyan[4], Ali Asghar Ghods[2*]

1 Department of Anesthesia, School of Allied Medical Sciences, Tehran University of Medical Sciences, Tehran, Iran, 2 Nursing Care Research Center, Semnan University of Medical Sciences, Semnan, Iran, 3 Critical Care Nursing Department, School of Nursing and Midwifery, Semnan University of Medical Sciences, Seman, Iran, 4 Research Center for Cancer Screening and Epidemiology and Health in Disaster and Emergencies Department, Nursing Faculty, Aja University of Medical Sciences, Tehran, Iran

* aaghods@yahoo.com

## Abstract

### Introduction

Innovative behavior is essential in nurses, driving continuous improvement and operational efficiency, significantly enhancing patient outcomes and overall healthcare quality. This study aims to evaluate the psychometric properties of a specific, theory-driven inventory for measuring nurses' innovative behavior in Iran.

### Materials and methods

A methodological study was conducted from November 2022 to April 2024. The conceptualization phase involved a qualitative study and a comprehensive literature review to define the concept of nurses' innovative behavior and identify its key dimensions. The subsequent psychometric evaluation assessed face validity, content validity, construct validity, and structural validity (using exploratory and confirmatory factor analyses) on a sample of 572 clinical nurses. Reliability was evaluated through internal consistency and the test-retest methods. Additionally, responsiveness and interpretability were examined according to the COSMIN checklist.

### Results

The construct validity of a five-factor structure (nurses' competencies, idea validation, clinical idea implementation, promoting innovation, and generating care ideas), identified during the conceptualization phase, was confirmed. Confirmatory factor analysis yielded a $\chi2/df$ ratio of 1.88 for the NIBI five-factor structure. The goodness of fit indices indicated suitable values (CFI=0.916, AGFI=0.817, IFI=0.917, PCFI=0.709, and RMSEA=0.057), with all factor loadings greater than 0.5 and statistically significant. Both convergent and divergent validities were demonstrated. The Cronbach's

---

---

**Data availability statement:** All relevant data are within the paper and its Supporting Information files.

**Funding:** The author(s) received no specific funding for this work.

**Competing interests:** The authors have declared that no competing interests exist.

alpha and omega coefficients ranged from 0.74 to 0.88 and 0.75 to 0.88, respectively. Additionally, the ICC for the entire inventory was 0.975 (CI 0.95–0.98, P < 0.001).

## Conclusion

The findings revealed that the Nurses' Innovative Behavior Inventory (NIBI) is both valid and reliable, making it a suitable tool for assessing and evaluating innovative behavior in nurses.

---

## 1. Introduction

Innovative behavior is characterized by the deliberate creation, advancement, and implementation of novel ideas, processes, or products in a position, team, or institutional setting [1]. This type of behavior involves not only creativity and the development of novel solutions but also the implementation of these solutions to improve efficiency, effectiveness, or quality in various settings [2]. In today's fast-paced and competitive environment, this behavior is essential for driving continuous improvement and adaptation, helping organizations remain relevant and successful [3].

In the rapidly evolving healthcare environment, innovative behavior among nurses is crucial for improving patient care, enhancing efficiency, and fostering a culture of continuous improvement [4]. Nurses' innovative behavior significantly impacts the healthcare system by driving improvements in patient care, operational efficiency, and overall healthcare quality [5]. Innovative behavior in nursing encompasses activities such as generating new ideas, implementing creative solutions, and promoting changes that enhance clinical practices and patient outcomes [6]. The existing literature on innovative behavior in nursing identifies several key factors that contribute to innovation. For instance, Asurakkody and Shin (2018) introduce organizational factors, work contextual factors, and personal factors as the antecedents of innovative behavior in nurses [7]. Other studies have explored the association among nurses' innovative behavior and various factors, including psychological capital [8,9], Leadership style [10], knowledge sharing, and job autonomy [11].

While several general innovation scales for measuring innovative behavior (IB) [12–14] and innovative work behavior (IWB) exist in the body of research [15,16], their application in the nursing field is limited due to the unique characteristics of this discipline. Importantly, creativity and innovative behavior, though related, are conceptually distinct: creativity refers to the generation of novel ideas, whereas innovative behavior encompasses the broader process of validating, implementing, and promoting those ideas within professional practice [17]. In nursing, this process must account for patient safety, ethical considerations, and interprofessional collaboration, which are often not captured in general instruments [7]. The need for a nursing-specific inventory is justified not only by the unique characteristics of the nursing discipline but also by the fact that our conceptual framework was empirically derived from the lived experiences of clinical nurses in Iran. This qualitative approach enabled us to identify critical dimensions that are highly relevant to clinical nursing

practice but absent in more general scales. Therefore, the present research aims to bridge this gap by evaluating the psychometric characteristics of a theoretically grounded instrument tailored to nurses' innovative behavior in Iran, developed through a combination of qualitative exploration and a comprehensive literature review.

## 1.2. Conceptual framework

The development of the Nurses' Innovative Behavior Inventory (NIBI) was guided by a conceptual framework derived from a conventional qualitative content analysis with clinical nurses in Iran and a comprehensive literature review. This two-part approach was used to define the concept of nurses' innovative behavior and identify its dimensions. The initial qualitative phase, aimed to deeply explore the experiences and perspectives of nurses in their specific context. The findings from this research revealed a five-dimensional structure of nurses' innovative behavior, which served as the conceptual foundation for the NIBI:

**Nurses' competencies:** This dimension is the foundation of innovation. It encompasses the essential skills and attributes nurses need, including critical thinking, creative self-efficacy, and a proactive mindset [18]. Modern practice also requires digital literacy to engage with new technologies, which are central to many innovations [19].

**Generating care ideas:** This is the creative phase where new concepts for improving care are born. Ideas often come from a nurse's direct observation of clinical problems and their ability to use divergent thinking [20]. Collaboration and a culture of curiosity are key to encouraging the free flow of ideas [21].

**Idea validation:** In this stage, an idea is assessed for its practicality and potential for success. The evaluation of feasibility is comprehensive, considering technical, economic, organizational, and ethical factors to ensure the innovation is viable and responsible [22].

**Clinical idea implementation:** This is the process of translating a viable idea into practice. It requires a structured approach, including planning, pilot testing, and gaining stakeholder buy-in [23]. This phase relies on effective communication and using implementation science principles to ensure the innovation is integrated successfully [24].

**Promoting innovation:** The final dimension is about creating an environment where innovation is continuously supported. Transformational leadership and providing structural empowerment are crucial for fostering a culture where nurses feel motivated and equipped to innovate [25]. This ensures that innovation is a sustainable part of nursing practice [26].

## 2. Design

This methodological research is derived from sequential exploratory mixed-method research from a nursing doctorate thesis conducted from November 2022 to April 2024. This study was conducted according to a protocol established by the authors [27].

## 3. Materials and methods

### 3.1. Conceptualization

In the initial phase, we a conventional qualitative content analysis using semi-structured interviews with clinical nurses (inductive method). This study adopted a qualitative descriptive design to explore the experiences and perspectives of nurses regarding their innovative behaviors. The data from the interviews were analyzed using a conventional content analysis approach [28] to systematically categorize qualitative data and uncover patterns and themes directly from participants' narratives. This process, which involved two researchers independently coding the transcripts, was followed by a collaborative and iterative grouping of codes into subcategories, and then into main categories that reflected the core dimensions of innovative behavior. Additionally, a comprehensive literature review (deductive method) was carried out to add any additional items to the item pool that were not identified in the interviews but were available in existing literature.

## 3.2. Validation process

The psychometric properties of the Nurses' Innovative Behavior Inventory (NIBI), which utilizes a Likert criterion with five points (spanning from 'Always' to 'Never'), were evaluated concerning face validity, content validity, construct validity, and reliability. The final 29-item English version of the Nurses' Innovative Behavior Inventory (NIBI) is provided as Supporting Information (S1 File). The original Persian version of the NIBI used for data collection is available in the Supporting Information (S2 File).

**3.2.1. Face validity.** The face validity of NIBI was assessed through both qualitative and quantitative approaches.

**Qualitative face validity assessment:** In this step 10 clinical nurses were recruited through convenience sampling and to review and provide feedback regarding the suitability, complexity, and clarity of the survey elements. The instrument was revised based on the participants' feedback.

**Quantitative face validity assessment:** To assess the face validity of the NIBI from a quantitative perspective, the item impact method was employed. The identical group of 10nurses who participated in the previous assessment of face validity were invited to rate the comprehensiveness of each item using a five-point rating scale ranging from 1 (Incomprehensible) to 5 (Fully comprehensible). The item impact score was determined by applying the calculation: comprehensiveness × Frequency (%). Within above calculation, frequency denotes the percentage of nurses rating the item as either 4 or 5, while comprehensibility refers to the average score derived from individuals' responses to that item. An impact score exceeding 1.5 signified that the item was deemed appropriate and was therefore kept in the measurement tool [29].

**3.2.2. Content validity.** According to COSMIN, content validity refers to the degree to which a scale's content precisely reflects the concept it is designed to assess. [30]. The content validity of NIBI was assessed through a combination of qualitative and quantitative approaches.

**Qualitative content validity assessment:** For the qualitative assessment of content validity, NIBI was reviewed by 15 experts (10 nursing doctorates and 5 specialists in scale development). These experts were asked to evaluate and provide feedback on the item allocation, wording, and scaling of the items. NIBI was revised based on their expert input.

**Quantitative content validity assessment:** To assess the necessity of the items a content validity ratio (CVR) was calculated based on the evaluations of 15 experts, using the threshold recommended in Lawshe's study [31]. When assessing CVR with 15 experts, according to the Lawshe's guidelines, the lowest required CVR value is 0.49. Items scoring below 0.49 may be excluded from the scale [31]. Twelve additional experts assessed the relevance of each item. The Content Validity Index for each item (I-CVI) were calculated. To account for the effect of chance, modified Kappa was calculated concerning every individual item according to the provided equation [32], with values ranging from 0.60 to 0.74 considered good, while values exceeding 0.74 considered excellent [33]. Furthermore, the scale-level content validity index (S-CVI) was assessed.

$$K = \frac{I.CVI - PC}{1 - PC} \rightarrow PC = \left[ \frac{N!}{A!(N-A)!} \right] \times 0.5^N$$

**3.2.3. Item analysis.** An item analysis was conducted within a pilot study involving 41 clinical nurses using convenience sampling with the goal of detecting possible concerns related to the items and computing inter-item correlations. Items with correlation coefficients below 0.3 were excluded. Additionally, it was examined whether removing the item would significantly enhance the internal consistency of the instrument [34].

**3.2.4. Construct and structural validity.** To assess construct validity, the factor structure of NIBI was evaluated using Principal Axis Factoring exploratory factor analysis (EFA), followed by Promax rotation, in SPSS version 26 [35]. The sample adequacy was assessed using the Kaiser-Meyer-Olkin (KMO) test, while Bartlett's sphericity test was used to evaluate the absence of the identity matrix. KMO values above 0.7 were considered acceptable [36]. Minimum acceptable factor loading was guided by following formula CV = 5.152 √ (n − 2), that yielded an approximate value of 0.32. In the given

equation, CV represents the number of factors that can be extracted, while n refers to the sample size [37]. Confirmatory factor analysis was employed to examine the structural factors, utilizing the maximum-likelihood approach and various commonly used goodness-of-fit measures. The evaluation of model fit was carried out based on root mean square error of approximation (RMSEA), parsimonious normed fit index (PNFI), parsimonious comparative fit index (PCFI), comparative fit index (CFI), incremental fit index (IFI), and chi-square/degree of freedom ratio (CMIN/DF).

The smallest required sample size for performing a factor analysis ranges from 5 to 10 participants per item of the instrument under investigation [38]. For this study, two independent samples were planned: one for exploratory factor analysis (EFA) and one for confirmatory factor analysis (CFA). Convenience sampling was employed to select the nurses. A sample of 310 nurses was used for Exploratory Factor Analysis, followed by a separate sample of 262 nurses for Confirmatory Factor Analysis. Participants eligible for inclusion were required to be clinical nurses with at least one year of experience and willing to participate. A questionnaire was designed in Porsline online survey platform, and the URL link was distributed via social media platforms (Telegram and WhatsApp) to groups of clinical nurses.

**3.2.5. Convergent and discriminant validity.** The convergent validity (the correlation among items within a single subscale) and discriminant validity (the distinctness of factors) of NIBI were assessed following the Fornell and Larcker's method [39] and the Heterotrait-Monotrait ratio of correlations (HTMT) method [40]. For convergent validity to be confirmed, the AVE must be higher than 0.5, and Composite Reliability (CR) should surpass 0.7. An HTMT ratio under 0.9 also suggests discriminant validity [40].

**3.2.6. Reliability, responsiveness, and interpretability.** The internal consistency of the Nurses' Innovative Behavior Inventory was assessed using Cronbach's alpha and Omega McDonald coefficients with scores exceeding 0.6 deemed satisfactory [41]. Test-retest reliability was examined by asking 40 clinical to complete the questionnaire on two separate occasions, with a three-week interval between the administrations using a two-way random effect. Responsiveness was evaluated using the standard error of measurement (SEM) and the minimal detectable change (MDC) values [42]. An MDC under 30% was deemed satisfactory, while an MDC of below 10% was supposed excellent [43]. The distribution of total scores, floor, and ceiling effects of NIBI was also assessed.

**3.2.7. Normality and outliers.** The dataset was assessed for univariate normality using skewness (±3), and multivariate normality was assessed through kurtosis (±7) [44].

The fully anonymized raw dataset underlying the analyses is available in the Supporting Information (S3 File).

## 3.3. Ethical consideration

This study, part of a nursing doctoral thesis, was approved by the ethics committee of Semnan University of Medical Sciences (CODE: IR.SEMUMS.1401.226). It aimed to develop a valid and reliable tool for measuring innovative behavior in clinical nurses. Surveys were conducted via the Porsline survey platform from 10th August 2023 to 8th November 2023. Written informed consent was obtained from participants, who agreed by selecting "I consent to take part." All personal data, including names, were anonymized for confidentiality.

## 4. Results

### 4.1. Conceptualization

During the conceptualization step of this research, 16 interviews were carried out, yielding a total of 1601 primary codes extracted through qualitative content analysis. Following further analysis, integration, or elimination of codes, the number was refined to 872 final codes. Based on the qualitative interview analysis, five main categories emerged, including Nurses' competencies, generating care ideas, feasibility of care ideas, clinical idea implementation, and promoting innovation, to conceptualize nurses' innovative behavior. Subsequently, an initial instrument comprising 50 items was designed, based on the findings of the qualitative study and an extensive review of existing research.

## 4.2. Validation process

In present study, most of the nurses were women, accounting for 59.2% and held a bachelor's degree (81.6%). A total of 221 participants (6.38%) were between the ages of 20 and 30 years, while the remaining were over 30 years old. Additionally, 69.5% of the participating nurses reported having more than 5 years of clinical work experience.

All 50 items had acceptable impact scores (>1.5). After CVR analysis, 12 items were removed, leaving 38 items. Following further refinement with Kappa values, 32 items were retained. The scale-level CVI/Ave was 0.889, indicating excellent content validity.

The Cronbach's alpha for the 32-item NIBI was 0.92. One item was removed due to corrected correlation scores below 0.3.

The study involved two independent samples: 310 participants for exploratory factor analysis (EFA) and 262 participants for confirmatory factor analysis (CFA). Overall, the questionnaire link was sent to 650 clinical nurses, with 572 completing the form, resulting in a response rate of 88%. The study experienced no missing data, as the surveys were administered through the Porsline online survey platform, which requires respondents to answer all questions sequentially, ensuring complete responses. In Principal Axis Factoring, the KMO test yielded a value of 0.922, and Bartlett's test resulted in a value of 4182.861 (p < .001). Two items were excluded because of their low factor loadings, during the exploratory factor analysis. The EFA (n = 310) revealed a 5-factor structure named nurses' competencies, idea validation, clinical idea implementation, promoting innovation, and generating care ideas. These five factors, comprising 29 items, explained 48.65% of the total variance in Nurses' innovative behavior among Iranian clinical nurses. Fig 1 depicts the scree plot in the factor analysis. Factor loadings in the rotated matrix are presented in Table 1.

The results from the Confirmatory Factor Analysis (n = 262) confirmed this structure with good fit indices ($\chi^2$/df = 1.88; CFI = 0.916; RMSEA = 0.057) (Table 2). All factor loadings exceeded 0.50 and were significant (p < 0.001). Fig 2 illustrates the Confirmatory Factor Analysis and the structural model for the NIBI.

Table 3 results indicate that the AVE of the factors was too close or surpassed 0.5, and the CR exceeded the AVE, meeting the criteria for convergent validity. Additionally, values in the HTMT matrix were below 0.9, signifying the establishment of discriminant validity in this study (Table 4).

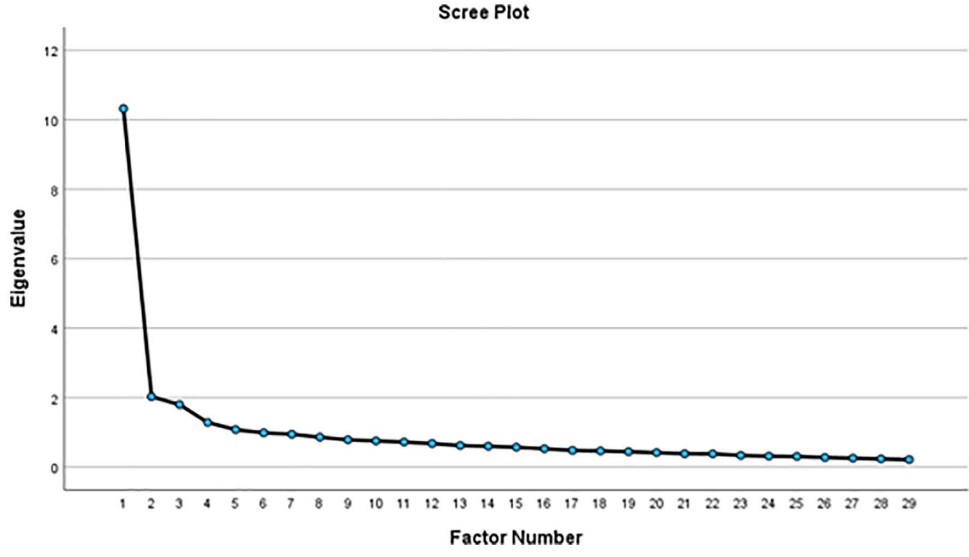

**Fig 1. Scree plot of NIBI.**

**Table 1. Factor loadings in the rotated matrix of the NIBI.**

| Items | Factor* | | | | |
|---|---|---|---|---|---|
| | 1 | 2 | 3 | 4 | 5 |
| I am looking to learn from online educational resources to enhance my nursing skills. | .804 | | | | |
| I increase my knowledge and skills by learning new techniques and methods. | .779 | | | | |
| I also learn essential non-nursing skills to create effective creativity in nursing work. | .707 | | | | |
| I am interested in solving nursing problems in an innovative way. | .690 | | | | |
| I seek to identify potential problems or inefficiencies in nursing. | .655 | | | | |
| I pay attention to emerging challenges in patient care and health care systems. | .492 | | | | |
| I use available scientific evidence to generate new ideas in patient care. | .403 | | | | |
| I have the ability to convince the supervisor and colleagues to accept my innovative idea. | .379 | | | | |
| I enjoy discovering new ideas and concepts in nursing. | .369 | | | | |
| I communicate effectively with the patient and her/his family. | .352 | | | | |
| I follow ethical standards and regulations when presenting innovative ideas. | | .920 | | | |
| When implementing an idea requires approval or permission, I seek guidance from informants. | | .812 | | | |
| When proposing and implementing a new idea, I consider the scope of my authority as a nurse. | | .789 | | | |
| When introducing my idea, I emphasize how this innovation can contribute to better patient care. | | .501 | | | |
| I consider the potential benefits and risks of a new idea for patients before implementing it. | | .413 | | | |
| When faced with unforeseen situations (such as errors in patient care), I think about why it happened. | | .338 | | | |
| I implement my care idea first for a limited number of patients. | | | .797 | | |
| To attract support, I run a simple model of my innovation. | | | .592 | | |
| I involve the patient's family and caregivers in the implementation of my innovative idea. | | | .568 | | |
| I use patients' feedback to refine my care ideas. | | | .537 | | |
| To implement my idea, I am looking for the participation of other nurses. | | | .513 | | |
| To implement my idea, I am looking for the participation of physicians and other healthcare workers. | | | .500 | | |
| I collaborate with universities to publish and share information about my innovation. | | | | .929 | |
| I participate in festivals or events to introduce and promote my innovation. | | | | .783 | |
| I share the details of my innovation with other nurses and physicians, so that it can be implemented elsewhere if needed. | | | | .409 | |
| I welcome innovative ideas from patients and their family caregivers to meet my patient's needs. | | | | | .807 |
| I am inspired by others' successful ideas to improve patient care in my department. | | | | | .571 |
| I find innovative solutions when faced with unforeseen obstacles in patient care. | | | | | .483 |
| To provide an innovative care idea, I anticipate the needs of each patient. | | | | | .413 |

*1: Nurses' competencies,2: Idea validation,3: Clinical idea implementation,4: Promoting innovation,5: Generating care ideas.

Extraction Method: Principal Axis Factoring. Rotation Method: Promax with Kaiser Normalization.

**Table 2. Fit model indices of confirmatory factor analysis of the NIBI (n = 262).**

| Indices | $\chi^2$ | p value | CMIN/DF | RMSEA | PCFI | AGFI | IFI | CFI |
|---|---|---|---|---|---|---|---|---|
| Values | 675.397 | <0.001 | 1.885 | 0.057 | 0.709 | 0.817 | 0.917 | 0.916 |

Cronbach's alpha and McDonald's omega for the five factors ranged from 0.74 to 0.88, indicating good to excellent internal consistency. Test–retest reliability was also excellent, with an ICC of 0.975 (95% CI: 0.95–0.98, p < 0.001) (See Table 3).

The percentage of minimum detectable change was 6.42%, with a Standard Error of Measurement of 2.57. The ceiling and floor effects in the inventory results were minimal, with a ceiling effect of 0% and a floor effect of 0.8%. ANOVA showed significant differences in innovative behavior scores by age group (p = 0.015) and education level (p = 0.048), with

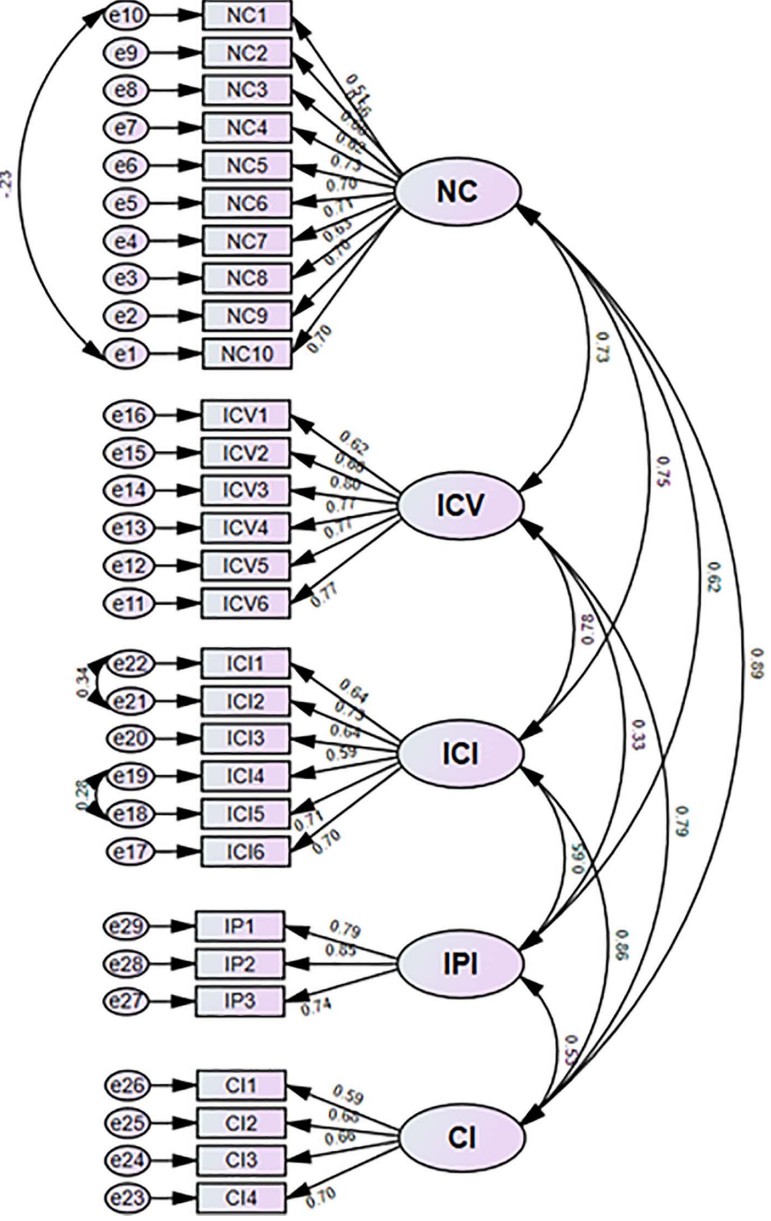

**Fig 2. The Nurses' Innovative Behavior construct: the model of confirmatory factor analysis (n = 262).** NC = Nurses' competencies, ICV = Idea validation, ICI = Clinical idea implementation, IP = Promoting innovation, CI = Generating care ideas.

**Table 3. Convergent validity and reliability parameters of the Nurses' Innovative Behavior construct.**

| Factors | CR | MaxR(H) | Alpha [CI 95%] | Omega | AVE |
|---|---|---|---|---|---|
| Nurses' competencies | 0.833 | 0.889 | 0.87 [0.76–0.93] | 0.88 | 0.432 |
| Idea validation | 0.875 | 0.883 | 0.96 [0.93–0.98] | 0.87 | 0.540 |
| Clinical idea implementation | 0.828 | 0.833 | 0.91 [0.83–0.95] | 0.84 | 0.447 |
| Promoting innovation | 0.836 | 0.846 | 0.97 [0.94–0.98] | 0.84 | 0.630 |
| Generating care ideas | 0.749 | 0.754 | 0.95 [0.91–0.97] | 0.75 | 0.429 |

**Table 4. HTMT of the Nurses' Innovative Behavior construct.**

| Factors | Generating care ideas | Promoting innovation | Clinical idea implementation | Idea validation | Nurses' competencies |
|---|---|---|---|---|---|
| Generating care ideas | | | | | |
| Promoting innovation | 0.836 | | | | |
| Clinical idea implementation | 0.797 | 0.746 | | | |
| Idea validation | 0.567 | 0.665 | 0.354 | | |
| Nurses' competencies | 0.887 | 0.742 | 0.746 | 0.642 | |

older nurses and those with higher education levels reporting higher scores. These findings support the instrument's sensitivity to known-group differences (Table 5).

## 5. Discussion

The purpose of this study was to develop and psychometrically evaluate the Nurses' Innovative Behavior Inventory (NIBI), a nursing-specific tool that captures the multidimensional nature of innovation in clinical practice. The results confirmed that the NIBI is valid, reliable, and theoretically robust, offering a framework that extends beyond existing measures of creativity and innovative work behavior.

Several instruments have been developed to assess innovative work behavior, most prominently those by Scott and Bruce (1994), Kleysen and Street (2001), and De Jong and Den Hartog (2010). While these scales have been widely applied across industries, they were not originally conceptualized for healthcare. Their primary focus is on idea generation [14], promotion, and realization [13,15], which, although important, do not fully capture the complex processes nurses navigate in clinical settings. By contrast, the NIBI introduces additional dimensions, such as idea validation and promoting innovation, that emerged from qualitative exploration with nurses and reflect the ethical, clinical, and organizational considerations of healthcare.

Recent efforts to design nursing-specific tools underscore this need. For instance, Cen et al. (2023) developed an instrument in Turkey addressing determinants of innovative behavior among nurses and midwives [45]. While our findings align with Can's study, indicating that the concept of nurses' innovative behavior is multidimensional, that instrument focused mainly on managerial, contextual, and individual predictors of innovation rather than the behaviors themselves. In comparison, the NIBI directly measures how nurses generate, validate, implement, and disseminate innovations in their daily practice, making it more actionable for assessing professional competencies. A research similarly identified idea generation, promotion of ideas, and application as dimensions of innovative behavior in nurses, which are consistent with the results of our study [7]. It could be argued that these three dimensions constitute the primary dimensions of innovative behavior across various fields. This assertion is supported by similar studies conducted in other domains, where these dimensions are often evident in some form or another [46–48].

**Table 5. Distribution of innovative behavior scores in clinical nurses.**

| Variables | Classifications | Mean±SD | Result |
|---|---|---|---|
| Age (year) | 20-30 | 108.58±18.748 | F=3.54 df=3 p=0.015 |
| | 31-40 | 111.38±15.373 | |
| | 41-50 | 116.06±17.557 | |
| | Over 50 | 121.55±10.373 | |
| Education levels | Bachelor | 110.90±17.066 | F=3.06 df=2 p=0.048 |
| | Master | 113.20±17.467 | |
| | PhD | 126.71±19.041 | |

The distinction between creativity and innovative behavior is particularly relevant in this context. Creativity relates to the generation of new ideas, whereas innovative behavior encompasses their evaluation, adaptation, and application in clinical care [17]. Existing creativity scales validated in Iran have provided useful insights into divergent thinking and idea generation but do not account for implementation or diffusion. The NIBI fills this gap by capturing the entire innovation process, from idea generation to organizational dissemination, thereby offering a comprehensive framework that is directly applicable to nursing practice.

Another contribution of this study is the methodological approach. The NIBI was developed through a sequential exploratory mixed-methods design, beginning with qualitative content analysis of nurses' lived experiences and complemented by a systematic review of the literature. This dual approach ensured that the instrument was both theoretically grounded and contextually relevant [49]. The subsequent psychometric evaluation followed international standards (COSMIN) and addressed multiple aspects of validity and reliability. High levels of Cronbach's alpha, McDonald's omega, and item correlations indicated that the five factors of the NIBI have acceptable internal consistency. Additionally, the results for composite reliability (CR) showed that the NIBI is reliable. One advantage of assessing CR is that it remains unaffected by the quantity of items or the derived structure; rather, it depends on the specific factor loadings of every latent variable [50]. NIBI also exhibited good convergent and discriminant validity. Importantly, the assessment also included responsiveness and interpretability, dimensions often neglected in scale development studies. These methodological choices strengthen confidence in the accuracy, stability, and practical usability of the NIBI across different nursing populations.

This study is not without limitations. First, reliance on self-report measures introduces potential bias. Second, the cross-sectional design limits conclusions about the stability of innovative behavior over time. Third, the study sample, while diverse, was limited to Iranian nurses, which may constrain generalizability. Future research should apply longitudinal designs to examine predictive validity, explore the influence of individual and organizational characteristics over time, and test the instrument using advanced psychometric methods such as Rasch analysis and structural equation modeling. Cross-cultural validation studies are also recommended to extend the applicability of the NIBI in international settings.

**Implications for Nursing and Health Policy:** The findings of this study highlight several key implications for nursing and health policy:

Integration into Professional Development Programs: The validated Nurses' Innovative Behavior Inventory (NIBI) provides a practical tool for identifying strengths and areas for improvement in innovative behavior among clinical nurses. Policymakers and healthcare leaders can incorporate the NIBI into professional development and training programs, aiming to foster innovation competencies, promote creative problem-solving, and support evidence-based care.

Advancing Nursing Education: Nursing educators can leverage the NIBI to design targeted curricula that enhance nurses' abilities to generate, validate, and implement new ideas. This tool ensures a structured approach to nurturing innovation, aligning with global efforts to prepare nurses for complex and rapidly changing healthcare environments.

Informing Health Policy Initiatives: By providing actionable insights into the innovative behaviors of nurses, the NIBI can inform policy initiatives aimed at enhancing workplace conditions that promote innovation. Policies supporting job autonomy, knowledge sharing, and access to resources for innovation could lead to significant improvements in healthcare quality.

Supporting Organizational Change: The dimensions identified in the NIBI, such as idea validation and promoting innovation, can serve as benchmarks for organizational assessments. Healthcare institutions can use these benchmarks to foster a culture of innovation, encouraging cross-disciplinary collaboration and the implementation of effective clinical practices.

Global Health Implications: Although developed in Iran, the NIBI's psychometric robustness makes it adaptable to diverse cultural and healthcare settings. This adaptability enables its use in international research and policy frameworks, contributing to the broader global discourse on nursing innovation and its role in enhancing patient care.

## 6. Conclusion

In conclusion, the NIBI represents a novel, valid, and reliable instrument for assessing innovative behavior in nurses. Unlike previously available tools, it is grounded in a nursing-specific conceptual framework, incorporates dimensions unique to clinical practice, and has undergone rigorous psychometric evaluation in line with COSMIN guidelines. While it was developed in Iran, the multidimensional framework of the NIBI makes it adaptable to other nursing contexts and potentially valuable for cross-cultural research. This study therefore adds both conceptual novelty and practical utility to the measurement of innovative behavior in nursing.

## Supporting information

**S1 File. Persian version of the Nurses' Innovative Behavior Inventory (NIBI).**
(DOCX)

**S2 File. English version of the Nurses' Innovative Behavior Inventory (NIBI).**
(DOCX)

**S3 File. Raw dataset (anonymized).**
(RAR)

## Acknowledgments

We sincerely thank the participants, research team, ethics committee, and administrators of Semnan University of Medical Sciences for their invaluable support in the development of this study.

## Author contributions

**Conceptualization:** Elham Shahidi Delshad, Mohsen Soleimani, Armin Zareiyan, Ali Asghar Ghods.

**Data curation:** Elham Shahidi Delshad, Armin Zareiyan.

**Formal analysis:** Elham Shahidi Delshad, Armin Zareiyan, Ali Asghar Ghods.

**Methodology:** Elham Shahidi Delshad, Mohsen Soleimani, Armin Zareiyan, Ali Asghar Ghods.

**Software:** Elham Shahidi Delshad, Armin Zareiyan.

**Supervision:** Ali Asghar Ghods.

**Validation:** Elham Shahidi Delshad, Mohsen Soleimani.

**Writing – original draft:** Elham Shahidi Delshad, Mohsen Soleimani, Armin Zareiyan, Ali Asghar Ghods.

**Writing – review & editing:** Elham Shahidi Delshad, Mohsen Soleimani, Armin Zareiyan, Ali Asghar Ghods.

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
