## [Decision Letter · Decision Letter 0]

8 Sep 2025

Dear Dr. Ghods,

Thank you for submitting your manuscript to PLOS ONE. After careful consideration, we feel that it has merit but does not fully meet PLOS ONE’s publication criteria as it currently stands. Therefore, we invite you to submit a revised version of the manuscript that addresses the points raised during the review process.

We look forward to receiving your revised manuscript.

Kind regards,

Noushin Kohan

Academic Editor

PLOS ONE

Journal Requirements:

2. Please amend the manuscript submission data (via Edit Submission) to include author Aliasghar Ghods

3. Please amend your authorship list in your manuscript file to include author Ali Asghar Ghods

5. Please remove all personal information, ensure that the data shared are in accordance with participant consent, and re-upload a fully anonymized data set.

Additional guidance on preparing raw data for publication can be found in our Data Policy (https://journals.plos.org/plosone/s/data-availability#loc-human-research-participant-data-and-other-sensitive-data) and in the following article: http://www.bmj.com/content/340/bmj.c181.long .

Additional Editor Comments:

Dear Authors,

To prevent unnecessary delays and expedite the review process, I would like to inform you that your manuscript is now ready for revision following the initial review stage. Upon thorough examination of the reviewers’ comments, it is important to highlight that Reviewer 2 has recommended rejection based on several significant and serious concerns. Given the critical nature of these comments, they represent an important opportunity to enhance the scientific quality of your manuscript and should therefore be addressed with full diligence and care.

Meanwhile, Reviewer 1 has provided constructive feedback which, although does not explicitly recommend rejection, points to several weaknesses that require correction and careful consideration.

We kindly ask you to carefully consider and comprehensively respond to each of Reviewer 2’s comments, implementing all necessary revisions in detail. At the same time, please also address the valuable points raised by Reviewer 1. Thoroughly addressing all reviewers’ remarks will facilitate a smoother and faster subsequent review.

After completing the revisions, please submit the revised manuscript along with a detailed, clear, and point-by-point response to each reviewer’s comments via the provided submission link.

Wishing you success in your revision,

Dr Nooshin Kohan

Academic Editor

Reviewers' comments:

Reviewer's Responses to Questions

**Comments to the Author**

1. Is the manuscript technically sound, and do the data support the conclusions?

Reviewer #1: Partly

Reviewer #2: No

2. Has the statistical analysis been performed appropriately and rigorously?

Reviewer #1: Yes

Reviewer #2: I Don't Know

3. Have the authors made all data underlying the findings in their manuscript fully available?

Reviewer #1: Yes

Reviewer #2: Yes

4. Is the manuscript presented in an intelligible fashion and written in standard English?

Reviewer #1: Yes

Reviewer #2: Yes

Reviewer #1: An interesting manuscript has been submitted. Identifying Nurses’ Innovative Behavior and strengthening these behaviors can lead to nurse satisfaction and increasing the quality of nursing services. The result can be the provision of quality nursing care.

The manuscript was reviewed. A few points are unclear to me:

The conceptual framework of the study should be written and the concepts in it should be explained.

What is the reason for choosing the title "Inventory"? Considering that a mixed method study with an exploratory approach was conducted, why was this title chosen?

In the first phase - Conceptualization, two qualitative studies was conducted. However, these are not mentioned in the abstract and only the psychometric method of the inventory is stated.

The method of Conceptualization needs further explanation. This step was carried out by conducting two qualitative studies, but the type of qualitative study was not stated and it is not clear how the item pool was completed by conducting this phase.

Although the psychometric stage of research instruments is carried out in several different and separate steps, it is necessary to mention such things as sample size, type of sampling, method, and expression of the most important results in each step. For example, in determining quantitative face validity, what was the sample size? Or in conducting Exploratory and Confirmatory Factor Analysis, how was the sample size selected and nurses divided into two groups? Also, the findings of the Construct Validity stage are precisely mentioned, but in other steps, the results are not well stated. The discussion needs to be rewritten. Considering the existing questionnaires on Innovative Behavior and the process of preparing and psychometrically validating each one, the discussion should be rewritten.

Reviewer #2: While your article has relatively few methodological flaws (for instance, some of the results are reported in the Methods section), your justification for the inadequacy of creativity assessment tools in nursing is insufficient. Creativity in nursing is not necessarily a culture- or society-specific issue; rather, it is primarily a professional competency, and numerous studies have been conducted on this topic. Standardized and widely accepted measurement tools already exist.In my opinion, this study lacks novelty. Furthermore, validated tools for measuring creativity among nurses are already available in the Iranian nursing community. Additionally, the Results section of your study is inadequate.

Sincerely,

**Do you want your identity to be public for this peer review?** For information about this choice, including consent withdrawal, please see our Privacy Policy

Reviewer #1: No

Reviewer #2: No

---

## [Author Response · Author response to Decision Letter 1]

30 Sep 2025

Dear prof. Noushin Kohan,

On behalf of all co-authors, I would like to sincerely thank you and the reviewers for the constructive and insightful comments provided on our manuscript. We carefully reviewed each point raised and have responded to all comments one by one. The reviewers’ feedback was highly valuable and has significantly improved the quality of our work. We are confident that the revised version is now much stronger than the original submission.

For clarity, we have prepared and uploaded a detailed response table indicating our replies to each comment along with the corresponding page numbers where revisions were made. In the manuscript file, the changes requested by Reviewer 1 are highlighted in yellow, and those requested by Reviewer 2 are highlighted in grey to facilitate the review process.

We greatly appreciate the time and effort invested by you and the reviewers, and we hope that the revised version meets the journal’s standards for publication.

With kind regards,

Ali asghar Ghods

Corresponding Author on behalf of all co-authors

---

## [Decision Letter · Decision Letter 1]

27 Nov 2025

The Nurses’ Innovative Behavior Inventory (NIBI): A Development and Validation Study

PONE-D-25-04660R1

Dear Dr. Ali Asghar Ghods

We’re pleased to inform you that your manuscript has been judged scientifically suitable for publication and will be formally accepted for publication once it meets all outstanding technical requirements.

Kind regards,

Maria José Nogueira, Ph.D.

Academic Editor

PLOS ONE

Additional Editor Comments (optional):

The article has been reviewed and is ready for publication.

Reviewers' comments:

Reviewer's Responses to Questions

**Comments to the Author**

Reviewer #3: All comments have been addressed

2. Is the manuscript technically sound, and do the data support the conclusions?

Reviewer #3: Yes

3. Has the statistical analysis been performed appropriately and rigorously?

Reviewer #3: Yes

4. Have the authors made all data underlying the findings in their manuscript fully available?

Reviewer #3: Yes

5. Is the manuscript presented in an intelligible fashion and written in standard English?

Reviewer #3: Yes

Reviewer #3: Thank you for the opportunity to review the manuscript titled “The Nurses’ Innovative Behavior Inventory (NIBI): A Development and Validation Study.” The study is well designed and follows an appropriate methodological structure for the development and psychometric validation of an instrument.

The statistical procedures applied are suitable. The use of separate samples for the exploratory and confirmatory factor analyses strengthens the robustness of the factorial structure. The reported fit indices are acceptable and the reliability coefficients demonstrate satisfactory internal consistency and temporal stability.

The authors have addressed the previous reviewers’ comments adequately and the manuscript has improved as a result of these revisions.

I have no further comments.

**Do you want your identity to be public for this peer review?** For information about this choice, including consent withdrawal, please see our Privacy Policy

Reviewer #3: No

---

## [Editor Report · Acceptance letter]

PONE-D-25-04660R1

PLOS One

Dear Dr. Ghods,

I'm pleased to inform you that your manuscript has been deemed suitable for publication in PLOS One. Congratulations! Your manuscript is now being handed over to our production team.

Kind regards,

on behalf of

Professor Maria José Nogueira

Academic Editor

PLOS One